**Data Availability Statement:** All relevant data are within the paper and its Supporting Information files.

**Funding:** This research was funded by the National Natural Science Foundation of China, grant number

# Impact of network density on the efficiency of innovation networks: An agent-based simulation study

**Lei Hua** [ID] [1,2] *, **Zhong Yang** [2], **Jiyou Shao** [2]

1 School of Management Science and Engineering, Nanjing University of Information Science and Technology, Nanjing, Jiangsu, China, 2 School of Business, Nanjing University, Nanjing, Jiangsu, China

\* hualei@nuist.edu.cn

## Abstract

Network density is an important attribute that affects the efficiency of innovation networks. However, the understanding of how network density affects the innovation efficiency of innovation networks is still unclear and even controversial. This paper uses a multiagent simulation method to study this problem. First, an innovation simulation model is established to describe the generation process of innovations in the context of network innovation, and a classical random network model is used to generate a test set of structures with different network densities. Then, the innovation model is run on the test set of networks to obtain the innovation efficiency of the structures with different network densities. The result shows that for explorative innovation, high network density is more conducive to improving innovation efficiency, and for exploitative innovation, low network density is more conducive to improving innovation efficiency. However, when network density is small enough to destroy network connectivity, it will lead to a large risk of innovation failure. Finally, the reasons for the results are further analyzed, and the theoretical and practical significance of the conclusions are discussed.

## 1 Introduction

The complexity and risk of innovation make innovation activities increasingly depend on collaboration among innovation subjects. Individuals form innovation networks with different structures through various economic and social connections, thus making innovation in the network context the main form of innovation. Firms frequently engage in bi- or multilateral cooperation to exchange knowledge, learn from each other and innovate [1]. As an innovation network is the carrier of the innovation process, it goes without saying that its structural configuration is likely to affect knowledge diffusion process at both the micro level and the systemic level [2], therefor has an important impact on innovation efficiency.

The question of what network structure is the most conducive to innovation has long been of great interest to scholars. Literatures from various disciplines have addressed this question through different perspectives of network change process [3], structural characteristics (e.g. core-periphery structures [4], degree distribution [5], and small-world network properties

71732002; China Postdoctoral Science Foundation, grant number 2020M671446. The funders had no role in study design, data collection and analysis, decision to publish, or preparation of the manuscript.

**Competing interests:** The authors have declared that no competing interests exist.

[6]), individual behavior process (e.g. broad casting [7, 8], free diffusion by random interactions [9, 10], and barter trade [9, 11]), network position of individuals [12–16], ability of actors affected by the knowledge diffusion process [17], and so on.

The agent-based modeling (ABM) approach is one of the most suited techniques to study this question. In ABMs, the agent is mainly characterized by a given set of goals and actions, and a given set of rules of social engagement, driving the interactions with other agents and the environment [18]. This allows investigation of the relationship between the behaviors of the agents at the individual level and the dynamics of the overall network at the systemic level [19].

The ABMs in this field often work in two interrelated ways: creating different network topologies and simulating the knowledge diffusion process within the networks. In this stream of research, there are many structurally distinct algorithms to construct different network topologies, such as the Erdös-Rényi (ER) random network [20], the Barabási-Albert (BA) network [5], the Watts-Strogatz (WS) network [6], and the Evolutionary (EV) network [2]. Each of them captures different characteristics of the corresponding network topology. There are also some different knowledge transferring mechanisms employed to simulate the interactions between the individuals within the network. For example, Cowan et al. [7] and Cowan and Jonard [11] employ the WS network to explore the impacts of structural characteristics of path lengths and clustering coefficient on the efficiency of knowledge diffusion processes with both broadcasting and barter trade mechanisms of knowledge flow. Cowan and Jonard [9] employ the BA network to explore the impacts of the structural holes on the efficiency of knowledge diffusion by random interaction process with both free diffusion and bater trade transactions. Hua et al. [10] employ the BA network algorithm to explore the impact of preferential attachment on the efficiency of innovation networks with a process of free diffusion by random interactions. Lovejoy and Sinha [8] construct a presentative test set of all possible structures with 10 nodes to explore the efficiency of different structures with both free diffusion by random interactions and broadcasting mechanisms of knowledge flow. Mueller et al. [2] employ four different structural algorithms (i.e., ER, BA, WS and EV) to explore the effect of structural disparities on barter trade knowledge diffusion process in networks, and highlight the relationship between degree distribution and network performance.

Among the many indicators used to measure network structure, network density is a considerably basic and important indicator. Network density refers to the number of connections between nodes within a network. Gnyawali and Madhavan [21] suggest that the number of network connections will greatly affect the communication and cooperation between individuals, so network density is an important factor affecting individual behaviors and effects. Jackson and Watts [22] also point out that individuals form and sever links connecting themselves to other individuals in evolving networks, and there is a clear difference between the behavior of individuals in networks with large densities and those in networks with smaller densities. More specifically, network density has an important impact on the diffusion process of the network. For instance, Nadini et al. [23] studied the epidemic spreading and vaccination strategies in an urban-like environment, and the result shows that the density of the network plays a critical role on the diffusion of both SIS and SIR epidemic processes. Also, changing the number of links has important influence on interconnection between the components of the system, which can be modeled as a percolation process. Percolation theory is an approach to study vulnerability of a system [24]. Callaway et al. [25] studied percolation on graphs with completely general degree distribution, giving exact solutions for a variety of cases, including site percolation, bond percolation, and models in which occupation probabilities depend on vertex degree.

Then, how does network density affect the efficiency of innovation networks? One view is that a greater density of the innovation network is more conducive to the generation of

innovation. Saxenian [26] conducted an in-depth study of innovation activities in Silicon Valley and suggested that a dense network is an important foundation for Silicon Valley enterprises to continuously generate innovation. Ahuja et al. [27] suggest that dense networks can promote innovation in a mature and stable environment. Rost [28] suggests that a high-density enterprise network is conducive to learning among enterprises and cluster innovation because a higher network density is conducive to obtaining useful information for enterprises. The other view is that sparse networks are more conducive to innovation. Burt [29] points out that low network density is conducive to the generation of innovation because high network density easily causes redundancy of information and knowledge inside the network, thus resulting in a low efficiency of knowledge flow. Cowan and Jonard [11] point out that efficient innovation networks often have small-world characteristics, and small-world networks themselves are a structure with low network density. According to Nerkar and Parachuri [30], an efficient innovation network consists of adding a few central nodes to a sparse network for pooling knowledge. Scholars have also tried to find a boundary between the two. Zhang et al. [31] indicate that network density will promote knowledge increase when it is high or low, while the inhibition of knowledge diffusion and knowledge innovation will appear when network density is moderate.

These viewpoints are of great significance for understanding the influence of network density on the innovation process, but previously published works have not yet reached a consensus on this issue. The differences are mainly caused by the following aspects. First, innovation efficiency is affected by many factors. Scholars start from different theoretical perspectives and adopt different research methods, and their conclusions are often based on specific research backgrounds, which lack universality to a certain extent. Second, innovations can be divided into different types. For example, innovation could be divided into explorative innovation and exploitative innovation from the perspective of ambidextrous innovation [32], and innovation networks show different knowledge dynamics under different types. The lack of clear differences between different types of innovation in research design is an important reason for the controversial conclusions of some studies. Third, various structural characteristics of innovation networks are often interrelated, and many research methods have difficulty separating the influence of network density on innovation from other influencing factors to a certain extent. Will et al. [33] noted that computational experiments are an effective method to solve this kind of problem, while Lovejoy and Sinha [8] adopted the method of network simulation to carry out this kind of research. Therefore, this study uses this research paradigm to explore this problem.

This paper uses a multiagent simulation method to study the effect of network density on innovation efficiency. First, from the perspective of knowledge management, an innovation simulation model is constructed to describe the process of innovation, and two kinds of innovations, namely, explorative and exploitative innovation, are distinguished by adjusting the parameter settings of the model. Second, a network test set with different densities is generated based on a famous random network model proposed by Erdös and Rényi [20]. Third, the innovation simulation model is run on different networks in the network test set to observe the efficiency of different kinds of innovations. Finally, the influence of network density on innovation efficiency is analyzed by adjusting the type of innovation and network density.

## 2 The innovation process model

### 2.1 The innovation process computational representation

According to Joseph Alois Schumpeter, innovation puts together existing elements into "new combinations", and this new combination should be able to be introduced into the production

system and create new values [34]. Based on this famous concept, scholars in the field of knowledge management define innovation as a new combination of existing knowledge [35, 36], and this new combination must be superior to the current situation and conform to the goal of innovation [37, 38]. Therefore, innovation is regarded as a process of searching, reorganizing, and selecting existing knowledge [39, 40]. Based on this idea, this paper constructs a model of the innovation process from the perspective of the knowledge dynamics of the innovation process.

First, assume that all the knowledge needed to realize a specific innovation constitutes a knowledge vector $L = \{k_1, k_2, \ldots k_l\}$. Each dimension represents a specific type of knowledge, and each value $k_i$ represents the amount of specific type of knowledge accumulated to achieve innovation. Without loss of generality, set $k_i = k$ in the simulation process.

Then, consider an innovation network consisting of $N$ nodes. Each node is defined by a vector $v_i = \{v_{i1}, v_{i2}, \ldots, v_{il}\}$ with the same dimension as $L$. The number of $v_i$ denotes the amount of knowledge required for innovation in the corresponding dimension. At the beginning, the knowledge needed by the innovation is distributed in different nodes in the network. Here to model this idea, we randomly select $l$ nodes, set their knowledge vectors as $\{k, 0, \ldots, 0\}$, $\{0, k, \ldots, 0\}$, $\ldots$, $\{0, 0, \ldots, k\}$ respectively. At the same time, set the knowledge vectors of other nodes all equal 0. As time goes by, the nodes communicate with each other within the constraints of the network structure and constantly update their knowledge vectors, thus generating "new combinations" of knowledge. This innovation occurs once there is a node in the network that integrates all the knowledge needed for innovation, in other words, its knowledge vector is identical to the innovation vector $L$.

Specifically, the communication rules between nodes are set as follows: at each time $t$, every node randomly selects another node from its neighbors in the network and communicates with it. The two interacting nodes, denoted by $i$ and $j$, randomly select a dimension $c$ as the current topic for discussion. If node $j$ has more knowledge than node $i$ on this dimension, that is, $v_{ic} < v_{jc}$, then node $i$ will learn from node $j$ through this conversation so that $v_{ic}$ will obtain an incremental $dv$. In contrast, $v_{jc}$ gets an incremental $dv$. With the limit of their ability, it is assumed that each node can communicate only once on a single topic at any given time. It means that each couple of nodes, at each time step $t$, can communicate only on a specific topic. Additionally, only idle nodes could be selected as partners. If node $i$ selects one of its neighbors $j$ to interact, then both $i$ and $j$ are occupied at this time step. Both of them are not able to interact with another of their neighbors until next time step $t+1$. If there is no idle node in the network, time $t$ ends.

We use the methodology proposed by Lovejoy and Sinha [8] to measure innovation efficiency. Let $NCTF$ (number of conversations to finish) denote the number of all the communications between all the nodes before innovation happens, which measures the innovation cost. Let $TTF$ (time to finish) denote the time consumed to generate the innovation, that is, the number of simulation steps from the initial state to the generation of the innovation, which measures the speed of innovation. In addition, if the knowledge in the network cannot be communicated adequately due to structural limitations, innovation may also fail to occur. This is usually caused by the lack of connectivity of the network structure and causes the failure of the knowledge required for innovation to be fully integrated. Let $PF$ (percentage of failures) denote the proportion of innovation failures among all the iterations of the simulation process with a specific network structure, which measures the risk of the innovation. Thus, an efficient innovation network refers to structures that can carry out innovation with less cost, less time, and a higher success rate.

## 2.2 Two different types of innovation

In addition, this model can distinguish between two different types of innovation, namely, explorative innovation and exploitative innovation. Exploration and exploitation are a pair of important concepts in organizational learning [41], innovation [42], entrepreneurship [43], and many other areas. March [44] divides the process of organizational learning into exploration and exploitation. He and Wong [45] draw from March's analytical framework and divide innovation into explorative innovation and exploitative innovation. Explorative innovation emphasizes acquiring new kinds of knowledge and strives to leave and go beyond the existing knowledge base [41], mainly through expanding the width of knowledge, which increases types of knowledge. The main feature of explorative innovation is that the innovation subject should collect a variety of different types of knowledge, and the amount of each type of knowledge is not necessarily large. Exploitative innovation emphasizes the extraction and improvement of existing knowledge [41], mainly through expanding the depth of knowledge, which is to increase the amount of knowledge of specific types. The main feature of exploitative innovation is that the innovation subject should accumulate an amount of knowledge, and the diversity of knowledge is less required. Benner and Tushman [46] pointed out that, on the one hand, enterprises need to acquire new knowledge, develop new products, and open new markets through explorative innovation. On the other hand, enterprises need to integrate existing knowledge and expand the functions of existing products through exploitative innovation to provide better services to customers in the existing market. In some cases, companies need to strike a balance between the two types of innovation, either by carrying out both types of innovation activities at the same time or by alternately adjusting the resources allocated to both types of innovation.

The setting of the innovation vector $L$ enables the model to distinguish the two different types of innovation. There are two important parameters in vector $L$, $l$ and $k$, which represent the number of knowledge types and the amount of knowledge in each type, respectively. When the value of $l$ is relatively large and the value of $k$ is relatively small, it means that there are many kinds of knowledge needed by this innovation, but the requirement of the amount of each kind of knowledge is not high. Innovation could be realized by searching and acquiring new kinds of knowledge. Therefore, this setting represents explorative innovation. In contrast, when the value of $l$ is relatively small and the value of $k$ is relatively large, there are fewer types of knowledge needed by the innovation but higher requirements for knowledge of specific types. Innovation could be realized through the accumulation and in-depth utilization of specific types of knowledge. Therefore, this setting represents exploitative innovation.

## 3 The test set of network structures

### 3.1 The test set based on the random network model

The Erdös-Rényi random network model was used to generate a network set for testing. Erdös and Rényi [20, 47] provide pioneering research on random networks, and here, we adopt one of their key models. The network generation rule of this model is described as follows: for a network composed of $N$ nodes, an edge is formed between every two nodes with a probability p, and the generation of any two edges is independent of each other.

When the size of the network is fixed, the value of probability $p$ determines the number of edges in the network, so it also determines the density of the network. Network density is usually defined as the ratio between the actual number of edges and the maximum possible number of edges in the network [48]. The maximum number of possible edges in a network of $N$ nodes is $C_N^2 = N(N-1)/2$. Assuming that the actual number of edges in the network

generated by this model under the control of probability $p$ is $m$, the network density is $2m/N$ $(N-1)$. Since $m$ is a random variable determined by $p$ and $N$, it is not difficult to prove that the expected value of network density is $E(2m/N(N-1)) = p$. Therefore, in the process of repeated simulation experiments, the probability $p$ can be used as the proxy for the network density. When $p = 0$, no edges are formed between any two nodes, and the actual number of edges in the network is 0, so the density of the network is also 0. When $p = 1$, there are edges connected between any two nodes in the network. In this case, the actual number of edges in the network is the maximum possible number of edges, so the density of the network is 1. Let the probability $p$ vary from 0 to 1. A group of random networks with different network densities can be obtained, which is the network test set adopted in this research.

The main reasons for selecting this network set as the test set of this research are as follows. First, the model realizes direct control of network density. Second, the influence of other network structure characteristics can be well controlled by this model. Generally, there is a high correlation between different network structure characteristics. However, in this model, the existences of any two edges are independent of each other. Therefore, the influence of other structural characteristics is eliminated through sufficient randomization, and the effect of network density is separated from many other factors.

### 3.2 Three important features of the test set

In this model, the network structure is only determined by two parameters, $N$ and $p$. There are already many known properties of this model, and three important conclusions that are related to the subsequent research in this paper are summarized as follows.

First, the initial linkage appears at the threshold of $1/N^2$ [49]. The network is likely to have no edges for $p$ less than $1/N^2$, while for $p$ greater than $1/N^2$, the probability of having at least one edge approaches 1.

Second, a loop appears at the threshold of $1/N$, accompanied by a large branch. When defining the size of a branch of a network, the convention is that a branch is called a large branch if it has at least $N^{2/3}$ nodes, and a large branch is the only large branch in the network. When $p$ is greater than $1/N$, the random network will have a large branch with a probability of approaching 1 [49].

Third, the network becomes connected at the threshold of $\log(N)/N$. One of the important discoveries of Erdös and Rényi is that random networks have a transition stage from connected to disconnected phases [47]. They show that when $p$ is greater than $\log(N)/N$, the network will be connected with a probability approaching 1, and when $p$ is less than $\log(N)/N$, the network will be disconnected with a probability approaching 1.

Interested readers will find further details on thresholds and phase transitions of Erdös and Rényi random network in the work of Jackson [49], where also presents the snapshots of corresponding structures near the different thresholds of p value. It will be useful in the discussion of the model results, while here we proceed with the operation of the model.

## 4 Parameter settings and simulation results

### 4.1 Parameter settings

The effect of network density on innovation efficiency can be observed when the innovation simulation model is run on the network test set. The following is the setting of experimental values used in the simulation process.

In the network model, only the network size needs to be preset. Let the network size be $N = 100$ throughout all the experiments. In the innovation model, the two parameters $k$ and $l$ involved in the innovation vector $L$, the knowledge increment $dv$ in the communication

process, and the knowledge distribution in the initial state need to be preset. The setting of the innovation vector $L$ determines the specific types of innovations. According to the foregoing analysis, in the case of explorative innovation, let $l = 20$ and $k = 1$, and then the innovation vector $L$ becomes a 20-dimensional vector $\{1, 1, \ldots, 1\}$. In the case of exploitative innovation, let $l = 2$ and $k = 20$, and then the innovation vector $L$ becomes a 2-dimensional vector $\{20,20\}$. In addition, without losing generality, let the knowledge increment in communications be $dv = 1$.

We used the simulation software NetLogo [50] to write the code and Origin [51] to carry out the data processing and graphic editing. Each experiment was repeated 10,000 times to produce sufficient statistical significance.

## 4.2 Results of explorative innovation

Figs 1 and 2 show the cost (*NCTF*), speed (*TTF*), and risk of failure (*PF*) of explorative innovations as a function of the network density controlled by variable $p$. The curve in the figure is the mean value of the corresponding index, and the line segment in the vertical direction represents the standard deviation at that point. For the convenience of analysis, two key positions of the $p$ value corresponding to the network size ($N = 100$) are also marked in the figures: $p = 1/N = 0.01$ and $p = \log(N)/N = 0.02$. According to the description of the structural characteristics of the random network in the previous section, when $p = 0.01$, loops and large branches begin to appear on the network. When $p = 0.02$, the global connectivity of the network begins to emerge. In Figs 1 and 2, both the metrics *NCTF* and *TTF* are not plotted for the entire interval of $p$, which means that the cost and the time before innovation goes to infinity, i.e., the system never reach innovation. This is coherent with the fact the *PF* is equal to 1, as this means that the innovation process fails in all of the 10000 times of repeated experiments.

According to Figs 1 and 2, both *NCTF* and *TTF* of explorative innovation decrease with the increase in $p$. In other words, for explorative innovation, the increase in network density is not

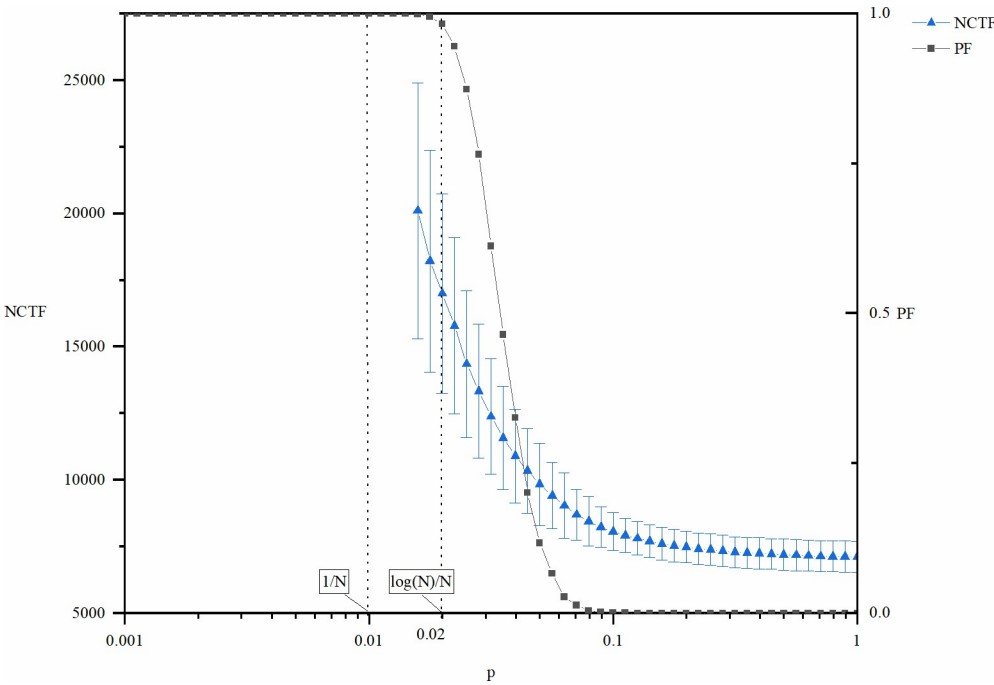

**Fig 1. NCTF and PF of explorative innovation as a function of p.**

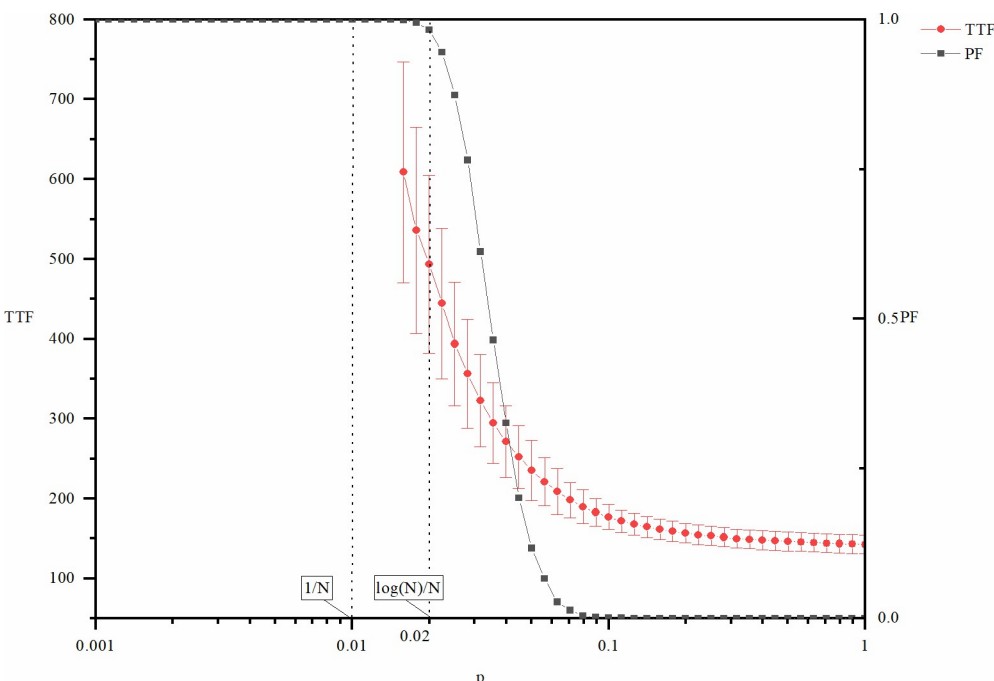

**Fig 2. TTF and PF of explorative innovation as a function of p.**

only beneficial to improve the innovation speed but also beneficial to reduce the innovation cost.

Figs 1 and 2 also show that explorative innovation will not occur if network density is remarkably low. This is because the connected branches of the network are relatively small, leading to the dispersed distribution of different types of knowledge required for innovation in different branches of the network so that the knowledge required for innovation cannot be fully realized. When the network density is remarkably high, the rate of innovation failure decreases because the connectivity of the whole network rises. Given enough time, the knowledge needed for innovation can be fully integrated. At the same time, the risk of failure has a state transition with increasing network density, and the reason behind this state transition is that the network structure changes from connected to disconnected stages with increasing network density [47].

In general, an increase in network density is beneficial to explorative innovation. Higher network density is beneficial not only to improve the speed of innovation and reduce the cost of innovation but also to avoid the risk of failure.

## 4.3 Results of exploitative innovation

Figs 3 and 4 show the cost (*NCTF*), speed (*TTF*), and risk (*PF*) of exploitative innovation as a function of the network density manipulated by *p*. Two key positions of *p* values are also indicated in the figures: $p = 1/N = 0.01$ and $p = \log (N)/N = 0.02$. In addition, another point $p = 0.05$ is also indicated in the figures. Both the *NCTF* and *TTF* curves present an inflection point here, especially that of the *TTF* curve, which is a local extreme point. Different from $p = 1/N = 0.01$ and $p = \log (N)/N = 0.02$, the position of this point depends not only on the structure parameters (*N* and *p*) but also on the parameters of the innovation model (*l* and *k*), so it cannot be expressed as a function of network size *N* with a relatively simple formula. This

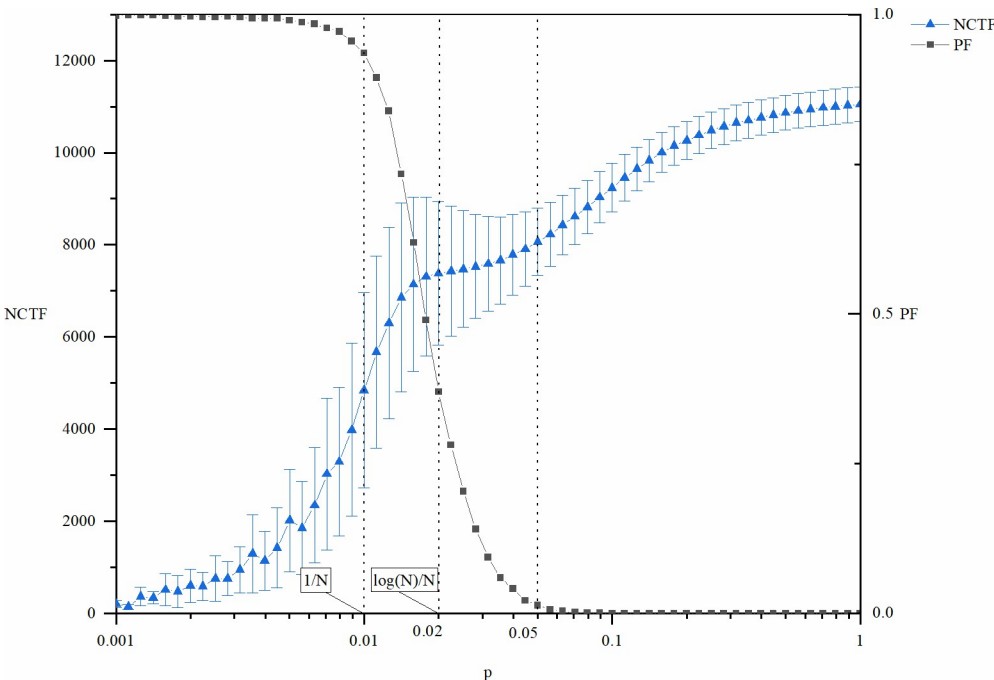

**Fig 3. NCTF and PF of exploitative innovation as a function of p.**

location is rather easy to identify both theoretically and practically. In this model, this point is bound to appear on the right side of $p = \log (N)/N$, where *PF* approaches zero. In other words, this is the critical point of global connectivity of the network. In management practice, participants of the network can easily determine whether they are linked to others.

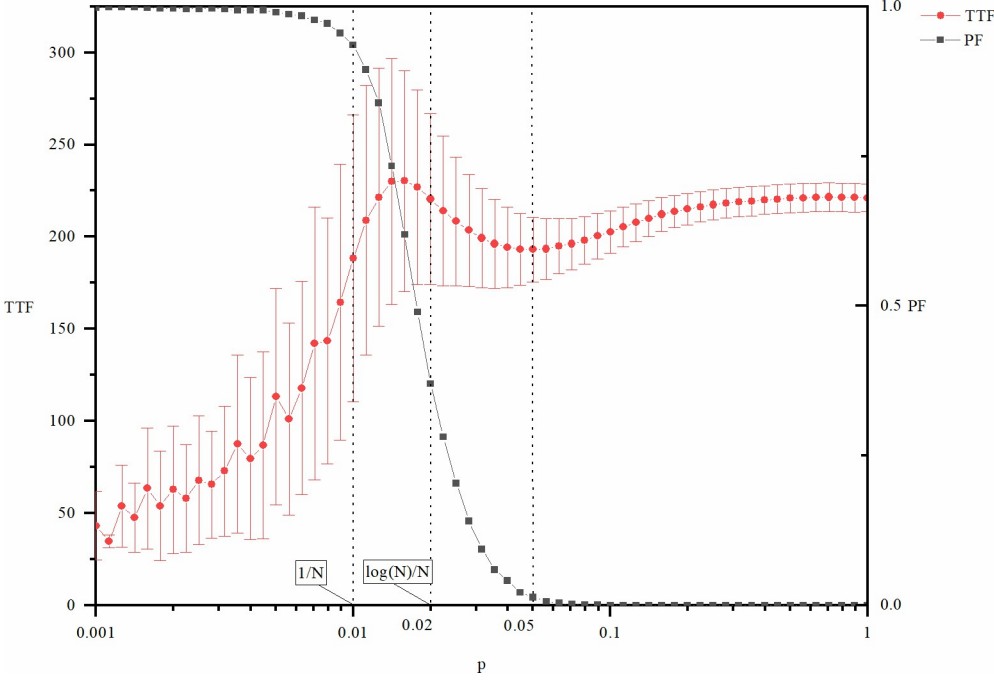

**Fig 4. TTF and PF of exploitative innovation as a function of p.**

As seen from Figs 3 and 4, the failure risk of exploitative innovation here has similar trends to that of explorative innovation shown in Figs 1 and 2, but there are also two differences. One is that even if $p$ is remarkably small, exploitative innovation can still occur, but explorative innovation cannot occur at this stage. Correspondingly, the value of $PF$ here is not equal to 1, although it looks very close to 1. This means that although the incidence is rather low, there are still a certain number of finite values of these metrics out of the 10000 repeated runs of the experiments. The second difference is that in exploitative innovation, the transition stage of innovation failure appears at smaller $p$. This suggests that the success of exploitative innovation is less dependent on network connectivity. Because exploitative innovation requires fewer types of knowledge, all the knowledge needed for innovation is provided by fewer participants.

The cost of exploitative innovation ($NCTF$) increases with increasing network density, as shown in Fig 3. Therefore, a lower network density is beneficial to reduce the cost of innovation. However, the $TTF$ of exploitative innovation changes in a more complex way. Although the changing mode of $TTF$ is similar to that of $NCTF$, it is no longer monotonous. Instead, there is an upward pulse in the middle part of the figure. When $p<0.01$ and $p>0.05$, $TTF$ of exploitative innovation increases with network density. Therefore, the lower the network density is, the faster the innovation process. However, when the network density is approximately $p = \log (N)/N = 0.02$, there is a special region that leads to poor innovation efficiency. The location of this special region is also determined by the transition stage of global connectivity of the network. A deeper explanation of this region will be provided below, along with further discussion of the mechanisms of the results.

In general, a high density of networks inhibits the efficiency of exploitative innovation. Higher network density not only leads to a higher cost of innovation but also leads to lower speed. A sparse network corresponds to a lower cost and faster speed, but when the network density is too low to maintain global connectivity of the network, the risk of failure of the innovation will increase sharply.

## 5 Further discussions on the simulation results

According to the three key points of the $p$ value, the discussion is divided into three scenarios. For clarity, we first discuss the scenarios of $p<0.01$ and $p>0.05$ and then discuss the scenario of $0.01<p<0.05$.

### 5.1 The scenario of $p < 0.01$

According to the proof in the literature [49], $p = 1/N = 0.01$ is the threshold of the emergence of loops and large branches in the network. When $p$ is less than this value, the network density is remarkably low, the network is made up of several small branches, and each small branch has a tree structure. With the increase in the $p$ value, the number of branches will gradually decrease, and the size of branches will gradually increase.

For explorative innovation, the generation process requires the integration of many different types of knowledge, so innovators need to communicate with many other subjects. Therefore, explorative innovation can only occur in large branches of the network. In such a structure of this scenario, there are hardly any large branches, so explorative innovation usually cannot occur.

For exploitative innovation, the generation process requires less knowledge. It is mainly realized by accumulating a few specific kinds of knowledge. What innovators need is not to communicate with many other subjects but to communicate repeatedly with a small number of individuals with needed knowledge. Therefore, exploitative innovation can occur in small branches of the network. However, due to the novelty of innovation, it is usually impossible to

know the exact location (which branch of the network) of needed knowledge, which leads to high failure risk in this scenario. If the needed knowledge exists in the same branch, innovation will potentially be realized. In such a circumstance, the smaller the branch is, the shorter the distance between needed knowledge. Then, innovation will be realized faster and at a lower cost.

## 5.2 The scenario of $p > 0.05$

In this scenario, the large branch of the network has absorbed most of the node, and the network structure has basically reached global connectivity. With the increase in $p$, the branch size has been basically fixed, so the risk of failure is extremely low, and innovation almost always occurs with a relatively high $p$ value.

In this case, the higher the network density is, the more neighbors each node has, which makes the communications between different nodes very convenient. Intuitively, the innovation network should be very efficient in this circumstance. However, the experimental results show that the increase in network density has opposite effects on the two different types of innovations. The concept of conversational churning emphasized by Lovejoy and Sinha [8] provides important enlightenment for understanding the reasons for these effects. Conversational churning refers to whether the node in the network can change communication objects frequently. A large network density means that the nodes in the network have many neighbors to talk with and results in a high degree of conversational churning.

For explorative innovation, high conversational churning can make the innovation subject have many communication objects to choose from, which will promote the integration of different types of knowledge to a large extent and improve the speed of innovation. At the same time, it can avoid wasting too much time in communication with the same partner, thus reducing the cost of innovation. Therefore, in this stage, the higher the network density is, the more conducive it is to the generation of explorative innovation.

For exploitative innovation, what the innovation subject needs is the accumulation of knowledge in specific fields. Frequent changes in communication objects will make the innovation subject unable to repeatedly discuss and learn about topics of a certain few areas, which will reduce the speed and increase the cost of innovation. Therefore, in this stage, the lower the network density is, the more conducive it is to the generation of exploitative innovation.

## 5.3 The scenario of $0.01 < p < 0.05$

In this scenario, when $p > 0.01$, loops and a single giant branch begin to appear in the network, and the number of loops and the size of giant branches gradually increase as the value of $p$ increases. At the same time, the threshold of global connectivity $p = \log(N)/N$ is also within this interval. Therefore, this interval reflects the phase transition of the network from unconnected to connected [47].

For explorative innovation, as innovation can only happen in large branches, innovations start to appear only when there are large branches in the network. Because of the uniqueness of large branches, the influence of network density on explorative innovation is relatively straightforward. As the size of a large branch increases, the probability of innovation failure ($PF$) drops rapidly. With increasing density, $TTF$ and $NCTF$ decrease, which means an increase in innovation efficiency.

For exploitative innovation, the possibility of innovation failure is slightly lower than that of explorative innovation because innovation is less constrained by the size of the branch where it happens. In this stage, the relationship between network density and innovation efficiency should be analyzed from the aspects of branch size and branch density.

According to the discussion of scenario $p<0.01$, when the network density is remarkably low, the branches in the network are mainly tree structures. Then, the size of the innovation branch determines the efficiency of exploitative innovation. The larger the size of the innovation branch is, the lower the innovation efficiency. According to the discussion of scenario $p>0.05$, when the network density is relatively high, innovation mainly occurs in only the giant branch, which has a small change in size. Then, the decisive role of branch size is replaced by branch density. The higher the branch density is, the lower the innovation efficiency will become. This scenario is in the middle of these two stages. As the loops started to appear and the tree structure started to diminish, there was a tradeoff between the effects of branch size and branch density.

To verify the above analysis, we trace the innovation branches that produce exploitative innovations in the numerical experiments. Let $BS$ denote the size of the innovation branch and $BL$ denote the number of edges in the innovation branch. The tree structure is the sparsest connected graph and the number of edges of any tree structure is equal to the number of nodes minus 1. $BL \geq BS-1$. Accordingly, let $BT = (BS-1)/BL$, then we have $0 < BT \leq 1$. Therefore, $BT$ can be used as a measure of the degree of the tree structure of the innovation branch. The larger the $BT$ value is, the more similar the branch is, such as to a tree. Finally, to capture the combined effect of the branch size and branch density, we multiply $BS$ and $BT$ to construct indicator $ST = BS(BS-1)/BL$. Fig 5 shows the size of the innovation branch $BS$, the number of edges in the innovation branch $BL$ and the structural index $ST$ as a function of $p$.

The results shown in Fig 5 support the discussion of the results. Compared with Figs 3–5 shows that in the $p<0.01$ scenario, the number of edges of innovation branches $BL$ changes little, and the innovation efficiency is mainly affected by the branch size $BS$ of the tree structure. When $p>0.05$, it is just the opposite. When $0.01<p<0.05$, the peak value of $ST$ explains the corresponding inflection point of the $NCTF$ and $TTF$ curves in Figs 3 and 4. At the same time, we can see that the influence of index $ST$ is stronger on $TTF$ (Fig 4) than on $NCTF$ (Fig 3).

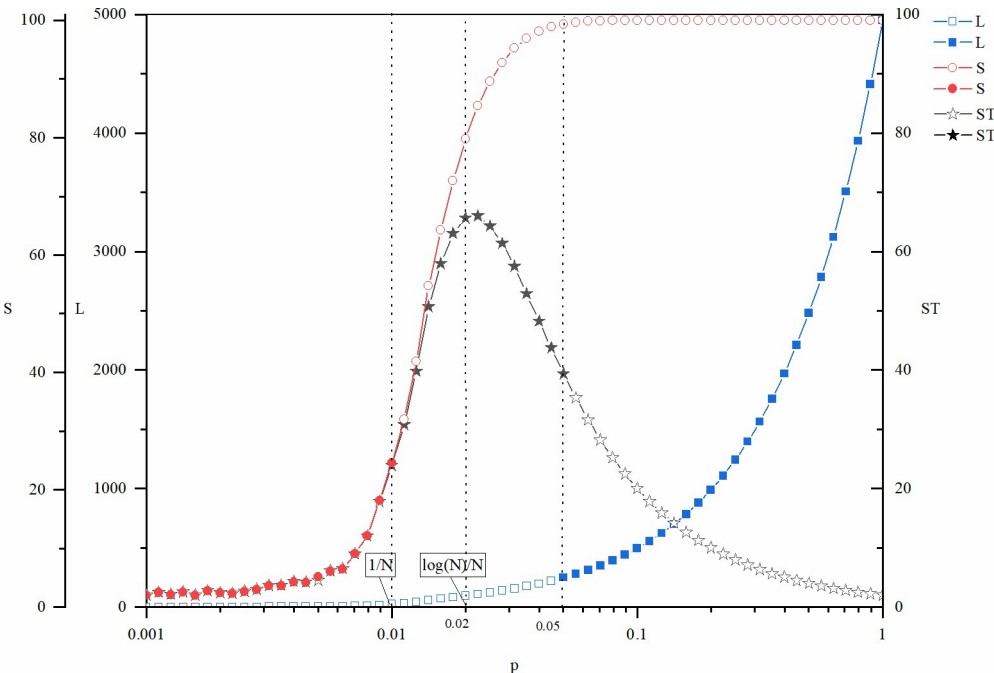

**Fig 5. Innovation branch structure characteristics of exploitative innovation.**

This is because the sparse nature of the tree structure greatly inhibits the redundant communications between nodes, which brings additional benefits to reduce the innovation cost, which reduces the rising trend of the *NCTF* curve with the rise of branch size.

## 6 Robustness tests

To test the robustness of the results, we adjust the values of all exogenous variables involved in the simulation model, mainly including the following three aspects.

First, other values of network model parameters.

For the network model used in this research, there is only one exogenous variable, namely, network size $N$. We run the model with network sizes of 50, 100, 150, and 200. The experimental results show that with an increasing network size, the three key points of the $p$ value will move towards the left in the original order. This is not difficult to understand because when $p$ is fixed, the expected number of edges in the network is $pN(N-1)/2$, resulting in the expected value of the average node degree being $p(N-1)$. The increase in node degree indicates the improvement of connectivity between different nodes. Therefore, when $p$ is fixed, the connectivity of the entire network will increase with the increase of $N$, resulting in the shift of the position where the structure of the network changes from disconnected to globally connected. However, this does not change the stage division according to the three key values of $p$, nor does it change the conclusions extracted from different stages. Fig 6 shows the model results with different values of network size $N$, while other parameters remain as the base case model.

Second, other values of innovation model parameters.

For the values of $l$ and $k$, different ratios determine the types of innovations. For explorative innovation, the cases with $l$ values of 10, 20, 40, and 80 are run under the condition of fixed $k$. For exploitative innovation, the cases with $k$ values of 10, 20, 40, and 80 are run under the condition of fixed $l$. Fig 7 shows the model results with different values of $k$ or $l$, while other parameters remain as the base case model. The experimental results show that the changes in $l$ and $k$ do not change the nature of the conclusions that are extracted from the base case model. Therefore, it gives a deeper understanding of the relationship between the innovation parameters and the probability of failure. As can be seen in Fig 7, the failure probability clearly depends on the size of knowledge vector $l$, while this does not depend on the amount of necessary knowledge $k$. This is because, in our model settings, innovation fails when the knowledge needed for innovation is distributed among different branches of a globally disconnected network, therefore the integration of different kinds of knowledge is ultimately impossible. For any specific structure of the network, a lager $l$ means that the innovation vector has more dimensions, and so it has more risk of being distributed in different disconnected branches. While the vector $k$ does not affect the number of dimensions of knowledge vector, and so it doesn't affect the risk of failure of each specific network structure.

Third, other knowledge flow mechanisms in communications.

In the process of innovation, when knowledge flows between two nodes, it is assumed that the knowledge increment of the node is $dv = 1$, which is consistent with the setting of the model of Lovejoy and Sinha [8]. Here, we also test the situation in which $dv$ is a certain proportion of the knowledge difference between the two nodes (see Fig 8) and when $dv$ varies with the knowledge level of the receivers (see Fig 9), while other parameters remain as the base case model. These results also do not change the nature of the conclusions that are extracted from the base case model. However, by inspecting Figs 1, 8 and 9, one can notice some differences in the values of both the *NCTF* and the *TTF*, for both explorative and exploitative innovation. On one hand, when the value of *PF* is very high, the ratio of successful innovation in the 10000 repetitions is reduced significantly, which decreases the stability of the results to some extent.

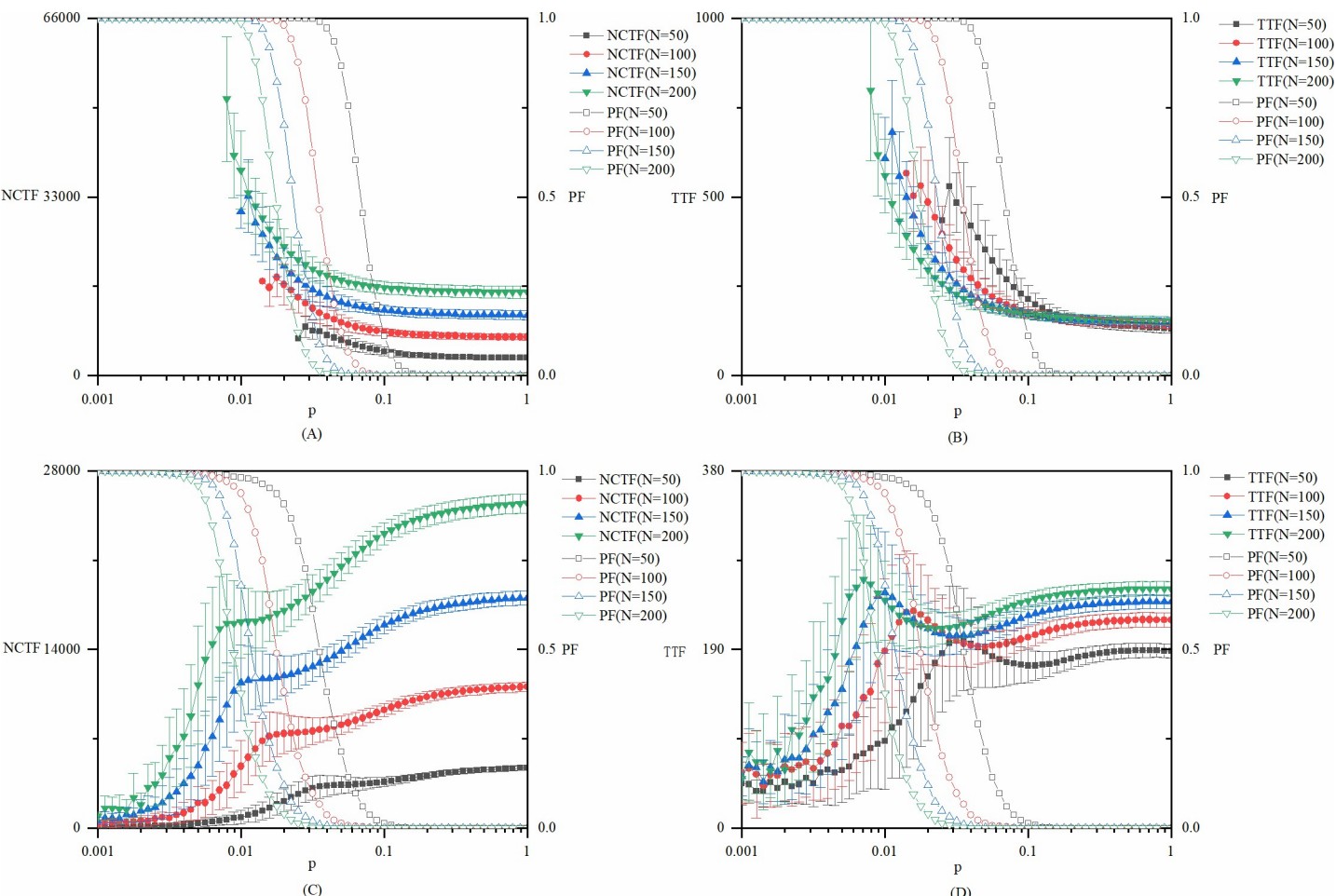

**Fig 6. Robustness test of the network size N.** (A) *NCTF* and *PF* of explorative innovations with different values of network size *N*. (B) *TTF* and *PF* of explorative innovations with different values of network size *N*. (C) *NCTF* and *PF* of exploitative innovations with different values of network size *N*. (D) *TTF* and *PF* of exploitative innovations with different values of network size *N*.

On the other hand, as the mechanisms of knowledge flow getting more complex in this part of the test, it also decreases the stability of the results. So, the left part of each curve (corresponding to low *p* value) shows a little more volatility than the base case model, but this doesn't change the overall trend of the curves. And as *p* value increases, all curves soon reach the expected stability and consistency.

All the changes not only retain the nature of the conclusions of the base case model, which indicates the robustness of this research in these tested aspects, but also bring some deeper understanding of the research question.

## 7 Discussions and conclusions

The understanding that many real-world systems of interacting elements can be mapped into graphs or networks has led to a surge of interest in the field of complex networks. When looking at their large-scale topological properties, real networks are far more complex than classical models of random graphs [52], and most of such models are rather abstract and often unable to reproduce the structure of real-world networks in all aspects. However, these models could capture some of the emerging properties not obvious at the level of their elementary

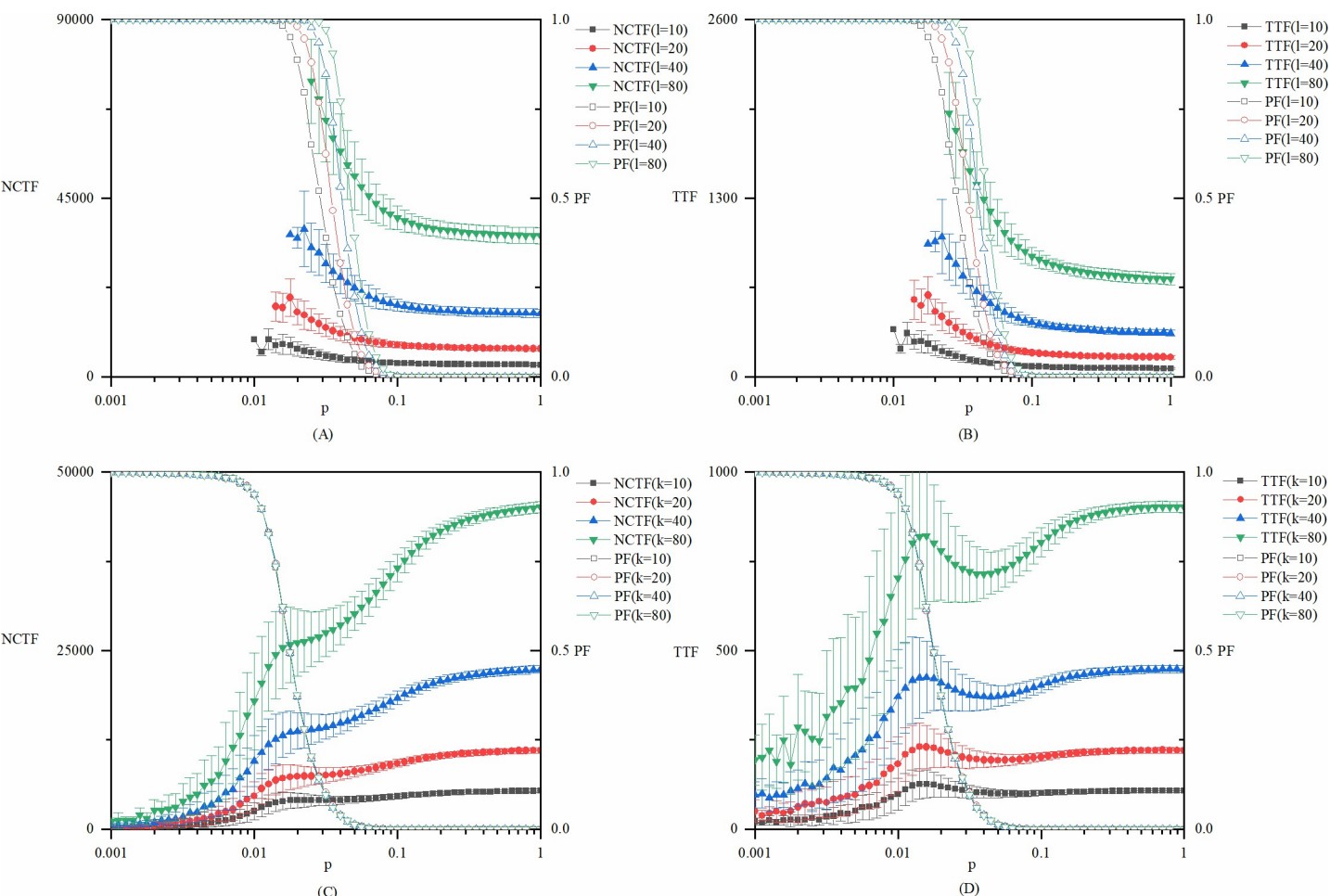

**Fig 7. Robustness test of the innovation model parameters k and l.** (A) *NCTF* and *PF* of explorative innovations with different values of *l*, which is the number of dimensions of the innovation vector *L*. (B) *TTF* and *PF* of explorative innovations with different values of *l*. (C) *NCTF* and *PF* of exploitative innovations with different values of *k*, which is the amount of knowledge in each dimension of the innovation vector *L*. (D) *TTF* and *PF* of exploitative innovations with different values of *k*.

constituents, such as the small-world effect, scale-free connectivity, clustering, degree correlations, etc., and further provide the opportunities to investigate their striking consequences on the behavior of the system, such as absence of epidemic threshold, resilience to damage, etc.

Based on this understanding, this paper is devoted to the in-depth study of the important property of network density, in particular its impact on the system behavior in the context of innovation networks. In order to separate the contribution of network density from other network features (e.g., degree heterogeneity, clustering, etc.), the Erdös-Rényi model is considered to generate the test set of different structures.

On a higher level of abstraction, this study contributes to the network diffusion literature by inquiring the relationship between network density and two different diffusion regimes: exploration, in which knowledge demands are shallow but broad; and exploitation, in which knowledge demands are deep but narrow. It is operationalized by controlling the number of types of knowledge and how much of each type is needed. This study also contributes to the literature on Erdös-Rényi networks by analyzing how the branch formation in the Erdös-Rényi model impacts the diffusion process, in particular the two regimes of exploration and exploitation.

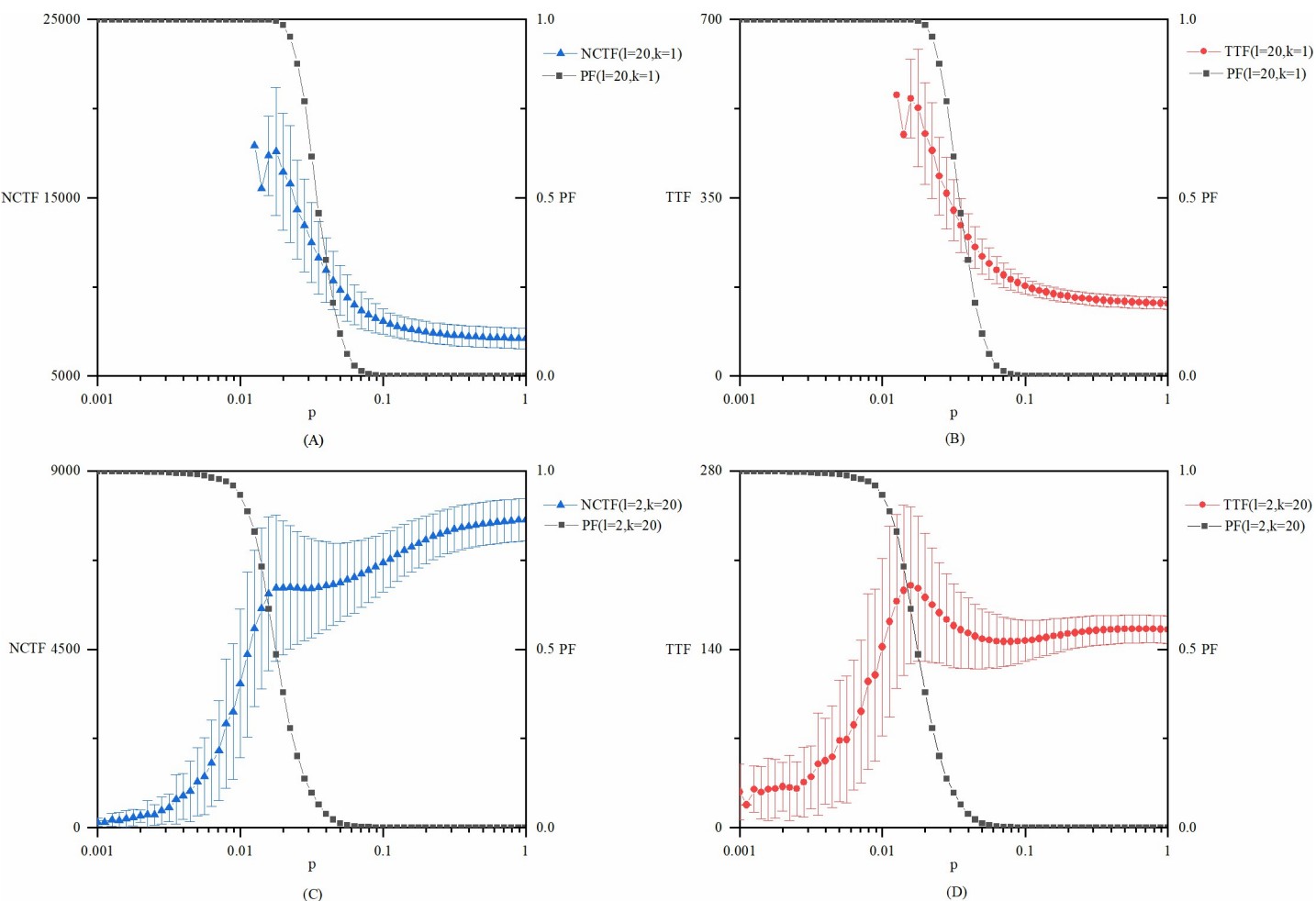

**Fig 8. The model results with dv as a certain proportion of the knowledge difference between the two nodes.** (A) *NCTF* and *PF* of explorative innovation. (B) *TTF* and *PF* of explorative innovation. (C) *NCTF* and *PF* of exploitative innovation. (D) *TTF* and *PF* of exploitative innovation. Specifically, the value of $dv = 0.2|v_i - v_j|$ is used in this experiment.

Specifically, this study contributes to the stream of literature on innovation networks by inquiring how network density affects the innovation efficiency of innovation networks. Extant research has shown equivocal evidences on this topic. This study analyzes the possible reasons behind the incoherent conclusions, and employs the multi-agent simulation method to disentangle this problem. First, in the context of network innovation, we build a simulation model to describe the generation process of innovation. Then, a network test is generated by a classical random network model, which consists of a group of random networks with different densities. By running the innovation model on the network test set, the innovation efficiency under different network densities is obtained. The conclusions derived from the model results shed light on the theory and practice of innovation management.

## 7.1 Main conclusions

For explorative innovation, the higher the network density is, the faster the innovation and the lower the innovation cost. Because explorative innovation requires the integration of many kinds of heterogeneous knowledge, it will occur on large branches of the networks. Due to the

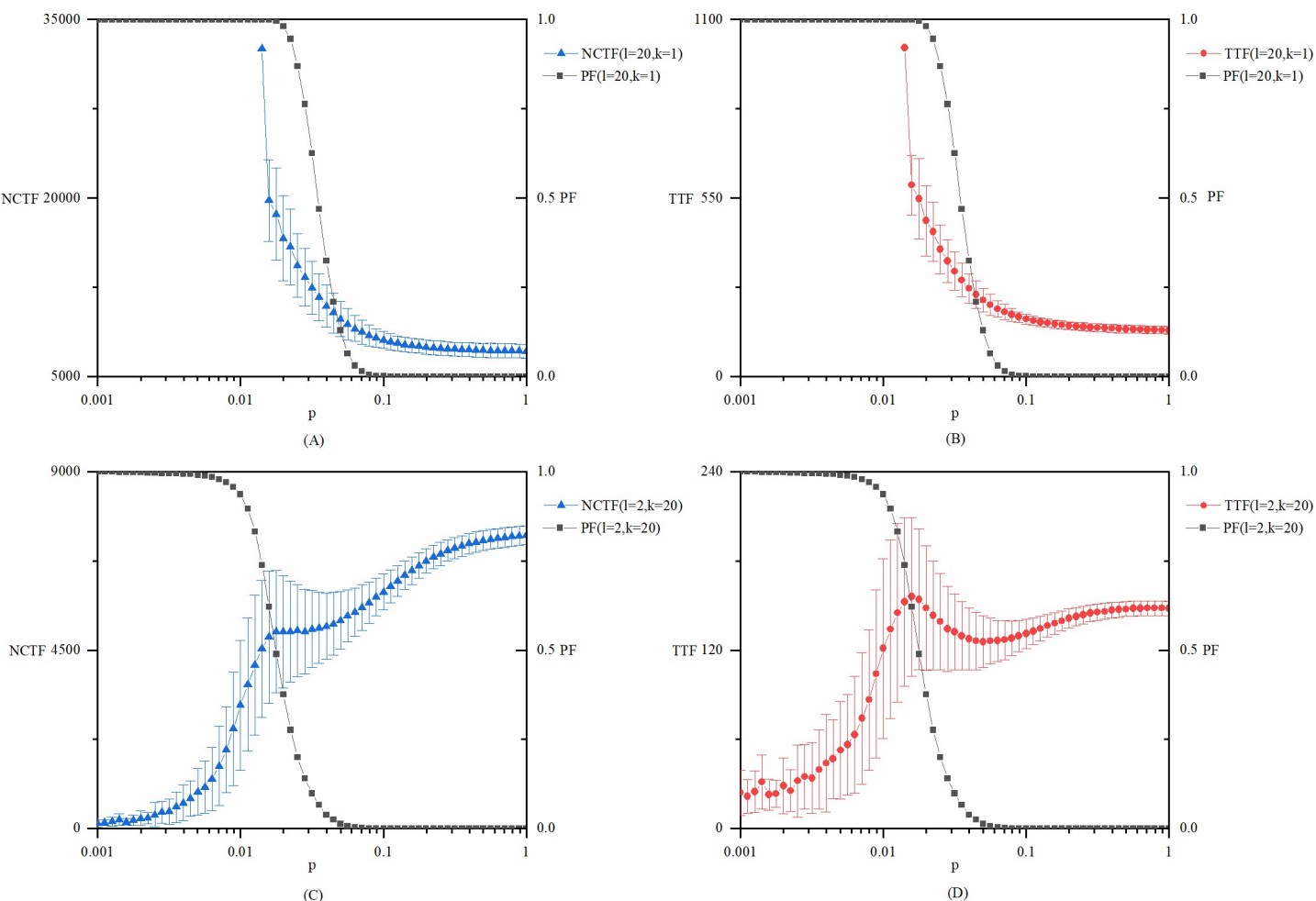

**Fig 9. The model results with dv varies with the knowledge level of the receivers.** (A) *NCTF* and *PF* of explorative innovation. (B) *TTF* and *PF* of explorative innovation. (C) *NCTF* and *PF* of exploitative innovation. (D) *TTF* and *PF* of exploitative innovation. Specifically, the value of $dv = 0.01(v_i−v_j)(v_j+1)$ is used in this experiment, as $v_j$ denotes the receiver of knowledge in this conversation.

uniqueness of the large branch of the random network, the change rule of innovation efficiency of explorative innovation is relatively straightforward, and both *NCTF* and *TTF* decrease monotonically with increasing network density. At the same time, the lack of global connectivity of the network is an important structural reason for the failure of innovation. If the network density is too low, there will be no large branches in the network, so explorative innovation cannot be produced. An increase in network density will inevitably lead to an improvement in the global connectivity of the network. High-density innovation networks are usually globally connected, so the risk of innovation failure is eliminated.

For exploitative innovation, low density is conducive to speeding up innovation speed and reducing innovation costs. However, since exploitative innovation requires fewer types of knowledge, it can be generated in both large and small branches. Therefore, compared with explorative innovation, the impact of network density on innovation efficiency is more complex. Therefore, we divide the analysis into three scenarios. First, when the network density is extremely low, there are only small branches in the network, and the smaller the branch size is, the higher the innovation speed and lower the innovation cost will be. However, the global connectivity of the network is poor, so the risk of failure is particularly high. Second, with the

increase in network density, large branches appear in the network and gradually transition to global connectivity. The decrease in tree structure, the increase in loops and the increase in branch size led to multiple inflection points in the *NCTF* and *TTF* curves in this process. In this scenario, with the increase in network density, the risk of innovation failure decreases rapidly, but the cost of innovation increases, and the time needed to generate innovation reaches a peak. Therefore, this interval is not conducive to the generation of exploitative innovation. Third, when network density increases to the point where global connectivity can be maintained, the efficiency of exploitative innovation decreases with increasing network density. This is because the conversational churning brought by the high network density makes the innovation subject unable to focus on the acquisition of specific kinds of knowledge.

## 7.2 Theoretical and practical significance

In terms of theoretical significance, this study clarifies the role of network density in the process of innovation. First, the multiagent simulation research method adopted in this paper has a higher level of abstraction, and the conclusions have more general applicability than empirical research. Second, this research uses the random network model to construct the test set, which realizes the direct manipulation of network density, and the influence of other characteristics of the network is well controlled by the randomization process. Third, the innovation model constructed in this paper distinguishes different types of innovation to identify the different influences of network density on different types of innovations. As a result, the conclusions of this research explained some controversial views in the extant literature.

In terms of practical significance, this study sheds light on the choice of innovation strategies of innovators and the formulation of innovation policies of innovation policy makers. Organizations pursuing explorative innovations should increase the density of innovation networks as much as possible, which is not only conducive to the improvement of innovation speed and cost but can also reduce the risk of innovation failure. For organizations pursuing exploitative innovation, a lower network density will bring a higher risk of failure, but a higher network density will lead to worse speed and cost. Therefore, a moderate network density would be a better choice at which level the entire network could just achieve global connectivity. This optimal point is not difficult to identify in management practice. When a globally connected network cannot maintain global connectivity by deleting a small number of edges or a disconnected sparse network can achieve global connectivity by adding a small number of edges, the network density is conducive to exploitative innovation. For the innovation subjects in each innovation network, it is usually not difficult to identify whether there are unreachable nodes in the network, so this conclusion is operable to a certain extent. In addition, if the participants of the innovation network can identify the key nodes with the knowledge required for innovation, it will greatly promote innovation efficiency and reduce innovation risks. Especially for exploitative innovation, it is easier and more important to identify the subject with expertise in a corresponding field because of the relatively few types of knowledge required and the need for a large amount of accumulation and in-depth development of specific kinds of knowledge.

## 7.3 Future research prospects

First, after the generation phase of innovations, organizations take other steps, such as implementation [53] and diffusion of innovations [54]. Therefore, in the subsequent stages of the innovation value chain [55], the role of network density remains to be further explored. Second, some other structural characteristics, such as the size or the degree distribution of the network, may have moderating effects on the results, which could be valuable of further

exploration to fit more specific scenarios with heterogeneous characteristics. Third, there are many other factors that affect innovation efficiency, such as resource endowment, institutional environment, organizational culture, and other aspects in addition to the structure of the innovation network. In this paper, the influence of network density on the innovation process is separated from many other factors through the method of experimental research. Although it contributes to innovation theory and practice, the interrelation among many factors should be considered in practice. How these factors interact with network density under different types of innovation needs further exploration. Fourth, more details in the conclusions of this research still need further verification by empirical research. The above aspects provide opportunities for further research but do not affect the current conclusions of this paper.

## Supporting information

**S1 File. The results of the base case model.** NCTF with N = 100, l = 20 and k = 1; TTF with N = 100, l = 20 and k = 1; NCTF with N = 100, l = 2 and k = 20; TTF with N = 100, l = 2 and k = 20; BL with N = 100, l = 2 and k = 20; BS with N = 100, l = 2 and k = 20.
(XLSX)

**S2 File. The results of robustness test on network size.** NCTF with N = 50, l = 20 and k = 1; NCTF with N = 100, l = 20 and k = 1; NCTF with N = 150, l = 20 and k = 1; NCTF with N = 200, l = 20 and k = 1; TTF with N = 50, l = 20 and k = 1; TTF with N = 100, l = 20 and k = 1; TTF with N = 150, l = 20 and k = 1; TTF with N = 200, l = 20 and k = 1; NCTF with N = 50, l = 2 and k = 20; NCTF with N = 100, l = 2 and k = 20; NCTF with N = 150, l = 2 and k = 20; NCTF with N = 200, l = 2 and k = 20; TTF with N = 50, l = 2 and k = 20; TTF with N = 100, l = 2 and k = 20; TTF with N = 150, l = 2 and k = 20; TTF with N = 200, l = 2 and k = 20.
(XLSX)

**S3 File. The results of robustness test on the parameters of innovation model.** NCTF with N = 100, l = 2 and k = 10; TTF with N = 100, l = 2 and k = 10; NCTF with N = 100, l = 2 and k = 20; TTF with N = 100, l = 2 and k = 20; NCTF with N = 100, l = 2 and k = 40; TTF with N = 100, l = 2 and k = 40; NCTF with N = 100, l = 2 and k = 80; TTF with N = 100, l = 2 and k = 80; NCTF with N = 100, l = 10 and k = 1; TTF with N = 100, l = 10 and k = 1; NCTF with N = 100, l = 20 and k = 1; TTF with N = 100, l = 20 and k = 1; NCTF with N = 100, l = 40 and k = 1; TTF with N = 100, l = 40 and k = 1; NCTF with N = 100, l = 80 and k = 1; TTF with N = 100, l = 80 and k = 1.
(XLSX)

**S4 File. The results of robustness test on different knowledge flow mechanisms.** NCTF with dv as a certain proportion of the knowledge difference between the two nodes and N = 100, l = 2, k = 20; TTF with dv as a certain proportion of the knowledge difference between the two nodes and N = 100, l = 2, k = 20; NCTF with dv varies with the knowledge level of the receivers and N = 100, l = 2, k = 20; TTF with dv as a certain proportion of the knowledge difference between the two nodes and N = 100, l = 2, k = 20; NCTF with dv as a certain proportion of the knowledge difference between the two nodes and N = 100, l = 20, k = 1; TTF with dv as a certain proportion of the knowledge difference between the two nodes and N = 100, l = 20, k = 1; NCTF with dv varies with the knowledge level of the receivers and N = 100, l = 20, k = 1; TTF with dv as a certain proportion of the knowledge difference between the two nodes and N = 100, l = 20, k = 1.
(XLSX)

## Acknowledgments

We would like to thank the editor and the reviewers for their comments, which assisted us in refining this paper.

## Author Contributions

**Conceptualization:** Lei Hua, Zhong Yang, Jiyou Shao.

**Data curation:** Lei Hua, Jiyou Shao.

**Methodology:** Lei Hua, Zhong Yang, Jiyou Shao.

**Software:** Lei Hua, Jiyou Shao.

**Writing – original draft:** Lei Hua, Zhong Yang, Jiyou Shao.

**Writing – review & editing:** Lei Hua.

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
