## [Decision Letter · Decision Letter 0]

7 Oct 2021

PONE-D-21-15597Impact of network density on the efficiency of innovation networks: An agent-based simulation studyPLOS ONE

Dear Dr. Hua,

Thank you for submitting your manuscript to PLOS ONE. After careful consideration, we feel that it has merit but does not fully meet PLOS ONE’s publication criteria as it currently stands. Therefore, we invite you to submit a revised version of the manuscript that addresses the points raised during the review process. Both reviewers ask to clarify on several points in the paper. Also, they raise issue on extendability of the approach beyond the ER network, and with real data sets. The authors are invited to discuss on these points. 

We look forward to receiving your revised manuscript.

Kind regards,

Alessandro Rizzo

Academic Editor

PLOS ONE

Journal Requirements:

Reviewers' comments:

Reviewer's Responses to Questions

**Comments to the Author**

1. Is the manuscript technically sound, and do the data support the conclusions?

Reviewer #1: Partly

Reviewer #2: Partly

2. Has the statistical analysis been performed appropriately and rigorously? 

Reviewer #1: N/A

Reviewer #2: Yes

3. Have the authors made all data underlying the findings in their manuscript fully available?

Reviewer #1: No

Reviewer #2: Yes

4. Is the manuscript presented in an intelligible fashion and written in standard English?

Reviewer #1: Yes

Reviewer #2: Yes

5. Review Comments to the Author

Reviewer #1: The article studies how the efficiency of innovation networks is modified by network density. Several Erdős and Rényi (ER) networks are generated and the efficiency of innovation studied and discussed. While I believe that the article might be a good addition to the present literature, I have several concerns and comments that should be first addressed

- More background literature should be added on the influence of network density on diffusion processes. For instance, network density influences the diffusion of disease processes on urban-like environments see: Nadini, M., Zino, L., Rizzo, A., & Porfiri, M. (2020). A multi-agent model to study epidemic spreading and vaccination strategies in an urban-like environment. Applied Network Science, 5(1), 1-30.

- Figures 1-4 are all about different snapshots of the same ER with increasing values of p. These figures should, in my opinion, be moved on the Appendix or even removed because ER networks are very well studied. You can refer to the proper literature (see network percolation) to highlight the influence of the parameter p on a network 

- The article does not study a realistic scenario and heterogeneity in degree distribution could be accounted. A way to easily generate arbitrary random networks is to follow: Perra, N., Gonçalves, B., Pastor-Satorras, R., & Vespignani, A. (2012). Activity driven modeling of time varying networks. Scientific reports, 2(1), 1-7. By aggregating several instances of a temporal network, a heterogeneous network is created. An even more popular approach is to use the configuration model to create random networks with heterogenous degree distribution. The Authors are free to explore and study different types of networks, which should encapsulate more realistic features

- Also, are results robust on variations of the size of the network N? For ER networks, there should be no problems at all in changing N, while for heterogeneous networks the parameter N may be an important factor.

- Could the Authors please provide more information about the data availability? Would the Authors share a folder with codes?

Reviewer #2: In the manuscript “Impact of network density on the efficiency of innovation networks: An agent-based simulation study”, the authors assess how the innovation process is affected by network density by using an agent-based modelling approach. In particular, the authors study different kinds of innovation, namely explorative and exploitative innovation, showing how network density have a different impact on the two, apparently solving the lack of consensus in the literature on the topic.

In general, I found the paper interesting. In particular, I really appreciated the analysis of how the branch formation in the Erdös-Renyi model impacts the innovation efficiency. However, the manuscript necessitates further improvement and some of the results need to be clarified.

First and foremost, the model needs to be better explained. In particular, two aspects are not completely clear to me. First, when the authors say, at line 110, “the knowledge needed by the innovation is distributed in different nodes in the network”, it means that when initializing the vectors v_i the vector L is “divided” between the network nodes? This aspect needs to be better clarified also in subsection 4.1, “Parameters setting”, as the initialization of the network is not described. Second, when the authors say, at line 122, “it is assumed that each node can communicate only once on a single topic at any given time”, does this mean that each couple of nodes, in each simulation, can communicate only on a specific topic? I think this mechanism is the crucial for understating the results the authors present. Additionally, assuming to consider a node A and to select one of its neighbors B to interact, will B successively interact with another of its neighbors? In general, it might be extremely useful for the comprehension of the article if the authors furnished the code or at least a pseudo-code for their model.

Second, some aspects of the results need to be better clarified. In Figs. 5 and 6, both the metrics NCTF and TTF have not been plotted for the entire interval of p. Does this mean that the cost and the time before innovation goes to infinity, i.e. the system never reach innovation? This would be coherent with the fact the PF is equal to 1, as this means that the failure of the innovation process is certain. However, by looking at Figs. 7 and 8, the cost and the time before innovations assume finite values even when PF=1. As these metrics, if I understood correctly, are finite only when the innovation occurs, how can their be so when PF=1? Such an aspect needs to be clarified. Additionally, when the network is globally connected and given the model rules, the two metrics should be related by the relation NCTF = N*TTF(/ 2 , if each node interacts only once at each time step). Is this the case?

Third, section 6, “Robustness tests”, needs to be improved. I would expect a figure or any other proof for each of the three aspects considered in the robustness analysis. Indeed, I think it is not correct to simply state that the experimental results show no significant changes without providing the outcome of these experiments.

Fourth, in order to study the impact of density and to separate its contribution from the one of other network features (e.g. degree heterogeneity, clustering, …) , the authors consider the Erdös-Renyi model. However, such a model is rather abstract and it is often unable to reproduce the structure of real-world network. Can the authors point out to which extent the characteristics of the real-world innovation networks can be reproduce by such a model? I believe such an analysis can better help to assess the practical implications of their study.

6. PLOS authors have the option to publish the peer review history of their article (what does this mean?). If published, this will include your full peer review and any attached files.

Reviewer #1: No

Reviewer #2: No

---

## [Author Response · Author response to Decision Letter 0]

10 Mar 2022

We would like to thank the reviewers for the kind, highly academic and helpful comments. It is of great help to improve our research. In particular, some of the comments not only help us to refine this article, but also made us more confident of the value of this research. We also thank the work of the editor to help improve this study.

 Now let us describe and reply to each comment one by one.

Response to Reviewer #1:

Comment (1):

 More background literature should be added on the influence of network density on diffusion processes. For instance, network density influences the diffusion of disease processes on urban-like environments see: Nadini, M., Zino, L., Rizzo, A., & Porfiri, M. (2020). A multi-agent model to study epidemic spreading and vaccination strategies in an urban-like environment. Applied Network Science, 5(1), 1-30.

Reply (1):

 We added more back ground literature on the influence of network density on diffusion process in the introduction (line 37-line 46), and added three references (literature [11-13]) accordingly, including the one given by the reviewer.

Comment (2):

 Figures 1-4 are all about different snapshots of the same ER with increasing values of p. These figures should, in my opinion, be moved on the Appendix or even removed because ER networks are very well studied. You can refer to the proper literature (see network percolation) to highlight the influence of the parameter p on a network 

Reply (2):

 We removed figures 1-4, and revised the corresponding narration (line 226-line 229). And we presented the literature to find corresponding details on thresholds and phase transitions of Erdős and Rényi random network for interested readers. 

We also added literature on percolation, and that helped to refine the review on network diffusion in the introduction (line 37-line 46). 

Comment (3):

 The article does not study a realistic scenario and heterogeneity in degree distribution could be accounted. A way to easily generate arbitrary random networks is to follow: Perra, N., Gonçalves, B., Pastor-Satorras, R., & Vespignani, A. (2012). Activity driven modeling of time varying networks. Scientific reports, 2(1), 1-7. By aggregating several instances of a temporal network, a heterogeneous network is created. An even more popular approach is to use the configuration model to create random networks with heterogenous degree distribution. The Authors are free to explore and study different types of networks, which should encapsulate more realistic features

Reply (3):

 We totally agree with the reviewer's suggestion, which will be very helpful to improve many of our studies. The given literature suggests a very good way to easily generate arbitrary random networks, and the configuration model could create random networks with heterogenous degree distribution, which could help us to encapsulate more realistic features. 

 In this paper, we focus on the important property of network density, in particular its impact on the system behavior in the context of innovation networks. In order to separate the contribution of network density from other network features (e.g., degree heterogeneity, clustering, etc.), the Erdős-Rényi model is considered to generate the test set of different structures. 

 As mentioned in extant literature, real networks are far more complex than classical models of random graphs, and most of such models are rather abstract and often unable to reproduce the structure of real-world networks in all aspects. Other parameters surely will have some influence on the experiments, and further the moderate the results of the model in this study. These suggestions point the way to further expand our findings in different areas in the future.

 With the help of this comment, we refined the discussions and conclusions (line 479-line 511), and the future research prospects (line 578-line 580). 

Comment (4):

 Also, are results robust on variations of the size of the network N? For ER networks, there should be no problems at all in changing N, while for heterogeneous networks the parameter N may be an important factor.

Reply (4):

 As the network density is defined as the ratio between the actual number of edges and the maximum possible number of edges in the network, and the maximum number of possible edges in a network of N nodes is C_N^2=N(N-1)/2, so it is vital important to test the robustness of the results on N. We added more evidence (see figure 6) in the section of robust test. It further confirmed that the results are robust on variations of the size of the network.

 As other features of the network, such as heterogeneity in some aspects (e.g., degree distribution, etc.) are not considered in this study. Yes, they may moderate the results in some extent, and that will be further explored in our future studies. 

Comment (5):

 Could the Authors please provide more information about the data availability? Would the Authors share a folder with codes?

Reply (5):

 We added some more details of the model to make it better explained. Especially on the initialization of the knowledge vector and some settings of the interaction between nodes during the simulation process. We tried to make it more convenient for the readers to be able to replicate the process described in the text. We also supplement the data sets supporting the main findings of this study. 

Response to Reviewer #2:

Comment (1):

 The model needs to be better explained in two aspects. 

 First, when the authors say, at line 110, “the knowledge needed by the innovation is distributed in different nodes in the network”, it means that when initializing the vectors v_i the vector L is “divided” between the network nodes? This aspect needs to be better clarified also in subsection 4.1, “Parameters setting”, as the initialization of the network is not described. 

 Second, when the authors say, at line 122, “it is assumed that each node can communicate only once on a single topic at any given time”, does this mean that each couple of nodes, in each simulation, can communicate only on a specific topic? I think this mechanism is the crucial for understating the results the authors present. 

 Additionally, assuming to consider a node A and to select one of its neighbors B to interact, will B successively interact with another of its neighbors? In general, it might be extremely useful for the comprehension of the article if the authors furnished the code or at least a pseudo-code for their model.

Reply (1):

 First, we added some explanation on how the knowledge of the innovation is distributed in different nodes at the start time: Here to model this idea, we randomly select l nodes, set their knowledge vectors as {k, 0, …, 0}, {0, k, …, 0}, …, {0, 0, …, k} respectively. At the same time, set the knowledge vectors of other nodes all equal 0 (line 112- line 114).

 Second, we added some explanation on how the nodes interacts in each time step (line 133-line 134). It means that each couple of nodes, at each time step (but not in each simulation experiment), can communicate only on a specific topic. 

 Additionally, we added explanation on how the nodes select their partners (line 126-line 130). Here if A selects B, then B will not be able to interact with others in this time step. If there is no idle node in the network, time t ends, and nest time step arrives, then both A and B will be able to select of be selected again. 

 By doing this, we made it more convenient for the readers to be able to replicate the process described in the text. We also supplement the data sets supporting the main findings of this study.

Comment (2):

 Second, some aspects of the results need to be better clarified. 

 In Figs. 5 and 6, both the metrics NCTF and TTF have not been plotted for the entire interval of p. Does this mean that the cost and the time before innovation goes to infinity, i.e. the system never reach innovation? This would be coherent with the fact the PF is equal to 1, as this means that the failure of the innovation process is certain. However, by looking at Figs. 7 and 8, the cost and the time before innovations assume finite values even when PF=1. As these metrics, if I understood correctly, are finite only when the innovation occurs, how can their be so when PF=1? Such an aspect needs to be clarified. 

Additionally, when the network is globally connected and given the model rules, the two metrics should be related by the relation NCTF = N*TTF(/ 2 , if each node interacts only once at each time step). Is this the case?

Reply (2):

 First, the numbers of Figs 5 and 6 are changed to Figs 1 and 2 in the new version of manuscript. We added explanation on these tow figures (line 257-line 261). The missing points of NCTF and TTF mean infinity, and none innovation occurs during all the 10000 times of repeated experiments at this point of p value.

 Second, PF in Figs 7 and 8 (Figs 3 and 4 in this version) are not equal to 1, but very near to 1. Every finite value of NCTF and TTF in the figures corresponds to a PF<1.

 Third, the relation between NCTF and TTF is determined by the specific structure of the network. Assume the network is globally connected. If it is a complete network (one link exists between any pair of nodes), the two metrics should be related by the relation NCTF=N*TTF/2. Otherwise, it might be different. For example, a star with 1 core and N-1 non-adjacent peripheral nodes will have the relation of NCTF=TTF. Because in each time step, only 1 conversation could take place, and as the core node is occupied, most peripheral nodes will have to wait. 

Comment (3):

 Third, section 6, “Robustness tests”, needs to be improved. I would expect a figure or any other proof for each of the three aspects considered in the robustness analysis. Indeed, I think it is not correct to simply state that the experimental results show no significant changes without providing the outcome of these experiments.

Reply (3):

 We added 4 figures as the proof of the three aspects of robustness tests. Also, we will upload all the date generated by these experiments. 

Comment (4):

 Fourth, in order to study the impact of density and to separate its contribution from the one of other network features (e.g. degree heterogeneity, clustering, …) , the authors consider the Erdös-Renyi model. However, such a model is rather abstract and it is often unable to reproduce the structure of real-world network. Can the authors point out to which extent the characteristics of the real-world innovation networks can be reproduce by such a model? I believe such an analysis can better help to assess the practical implications of their study.

Reply (4):

 This comment is very helpful to refine the discussion and conclusions of this paper. We revised the corresponding part of line 479-line 511. 

As mentioned in extant literature, real networks are far more complex than classical models of random graphs, and most of such models are rather abstract and often unable to reproduce the structure of real-world networks in all aspects. However, these models could capture some of the emerging properties not obvious at the level of their elementary constituents, and further provide the opportunities to investigate their striking consequences on the behavior of the system.

 Our model focus on the property of network density and contributes to the network diffusion literature by inquiring the relationship between network density and two different diffusion regimes: exploration and exploitation. And we also contributes to the literature on Erdős-Rényi networks by analyzing how the branch formation in the Erdős-Rényi model impacts the diffusion process. 

 Specifically, this study contributes to the stream of literature on innovation networks by inquiring how network density affects the innovation efficiency of innovation networks. Extant research has shown equivocal evidences on this topic. This study analyzes the possible reasons behind the incoherent conclusions, and employs the multi-agent simulation method to disentangle this problem. The conclusions derived from the model results shed light on the theory and practice of innovation management, including the choice of innovation strategies of innovators and the formulation of innovation policies of innovation policy makers.

---

## [Decision Letter · Decision Letter 1]

11 Apr 2022

PONE-D-21-15597R1Impact of network density on the efficiency of innovation networks: An agent-based simulation studyPLOS ONE

Dear Dr. Hua,

Thank you for submitting your manuscript to PLOS ONE. After careful consideration, we feel that it has merit but does not fully meet PLOS ONE’s publication criteria as it currently stands. Therefore, we invite you to submit a revised version of the manuscript that addresses the points raised during the review process. In particular, while a reviewer suggests some minor modifications, another one echoes the reviewer in the first round by requiring a more in-depth analysis of the literature on agent-based models. We agree that this is a crucial point that has to be addressed. 

We look forward to receiving your revised manuscript.

Kind regards,

Alessandro Rizzo

Academic Editor

PLOS ONE

Reviewers' comments:

Reviewer's Responses to Questions

**Comments to the Author**

1. If the authors have adequately addressed your comments raised in a previous round of review and you feel that this manuscript is now acceptable for publication, you may indicate that here to bypass the “Comments to the Author” section, enter your conflict of interest statement in the “Confidential to Editor” section, and submit your "Accept" recommendation.

Reviewer #2: All comments have been addressed

Reviewer #3: (No Response)

2. Is the manuscript technically sound, and do the data support the conclusions?

Reviewer #2: Yes

Reviewer #3: Yes

3. Has the statistical analysis been performed appropriately and rigorously? 

Reviewer #2: Yes

Reviewer #3: N/A

4. Have the authors made all data underlying the findings in their manuscript fully available?

Reviewer #2: No

Reviewer #3: Yes

5. Is the manuscript presented in an intelligible fashion and written in standard English?

Reviewer #2: Yes

Reviewer #3: Yes

6. Review Comments to the Author

Reviewer #2: The authors have addressed all my comments and I believe that, in its current form, the manuscript has improved. However, there are still some issues that are worth tackling.

1) On lines 456-457, the authors comment Fig. 7 stating that “The experimental results show that the change in l and k does not change the nature of the conclusions.” However, I think it would be worth further discussing the results displayed in such a figure. For instance, in the case of explorative innovation, the failure probability clearly depends on the size of knowledge vector l, while, for the exploitative innovation, this does not depend on the amount of necessary knowledge k. I think the authors should further comment on this outcome.

2) When discussing the robustness of their multiagent model to a variation in the “knowledge flow mechanisms in communications”, on lines 465-483, the authors simply claim that the nature of the conclusion remains unchanged. However, by inspecting Figs. 1, 8, and 9, one can notice some differences in the values of both the NCTF and the TTF, for both explorative and exploitative innovation. I think this result should be discussed in a revised version of the manuscript.

3) While it is extremely useful that the authors have provided the data generated from their numerical analysis, I believe that the multiagent simulation code should be made as well available. Indeed, since the model characteristics are central to the manuscript, I retain the availability of the code to be crucial.

Reviewer #3: The article deals with the questions to what extent network density regulates the diffusion of innovation. As such, the research question is relevant and pertinent. However, as already mentioned by the reviewers of the first round, I first recommend conducting a deeper analysis of the existing literature in this field with a particular focus on agent-based models, such as Mueller, M., Bogner, K., Buchmann, T. et al. The effect of structural disparities on knowledge diffusion in networks: an agent-based simulation model. J Econ Interact Coord 12, 613–634 (2017). https://doi.org/10.1007/s11403-016-0178-8.

7. PLOS authors have the option to publish the peer review history of their article (what does this mean?). If published, this will include your full peer review and any attached files.

Reviewer #2: **Yes: **Luca Gallo

Reviewer #3: No

---

## [Author Response · Author response to Decision Letter 1]

11 May 2022

Response to Reviewers

 We would like to thank the editors and reviewers for their valuable comments and suggestions. We respond to every suggestion made by the editors and the reviewers in this revision. In particular, we attach great importance to the opinion of requiring a more in-depth analysis of the literature on agent-based models.

 Now let us describe and reply to each comment one by one.

Response to Reviewer #2:

Comment (1):

 On lines 456-457, the authors comment Fig. 7 stating that “The experimental results show that the change in l and k does not change the nature of the conclusions.” However, I think it would be worth further discussing the results displayed in such a figure. For instance, in the case of explorative innovation, the failure probability clearly depends on the size of knowledge vector l, while, for the exploitative innovation, this does not depend on the amount of necessary knowledge k. I think the authors should further comment on this outcome.

Reply (1):

 We further discussed the results in Fig 7 (lines 481-494).

 Here we addressed the fact that “in the case of explorative innovation, the failure probability clearly depends on the size of knowledge vector l, while, for the exploitative innovation, this does not depend on the amount of necessary knowledge k”. 

 We also discussed the reasons of this outcome. And it is also shown that this result doesn’t change the conclusions extracted from the base case model, which are mainly about the relationships between the density of the network (p) and the parameters of innovation efficiency (TTF and NCTF). 

Comment (2):

 When discussing the robustness of their multiagent model to a variation in the “knowledge flow mechanisms in communications”, on lines 465-483, the authors simply claim that the nature of the conclusion remains unchanged. However, by inspecting Figs. 1, 8, and 9, one can notice some differences in the values of both the NCTF and the TTF, for both explorative and exploitative innovation. I think this result should be discussed in a revised version of the manuscript.

Reply (2):

 We gave further discussions on this point (lines 507-516).

 Here we addressed this issue, explained the reason, and reconfirmed robustness of the results of the base case model.

Comment (3):

 While it is extremely useful that the authors have provided the data generated from their numerical analysis, I believe that the multiagent simulation code should be made as well available. Indeed, since the model characteristics are central to the manuscript, I retain the availability of the code to be crucial.

Reply (3):

 We upload the program file of our base case model to the submission website for review. 

Response to Reviewer #3:

Comment:

 The article deals with the questions to what extent network density regulates the diffusion of innovation. As such, the research question is relevant and pertinent. However, as already mentioned by the reviewers of the first round, I first recommend conducting a deeper analysis of the existing literature in this field with a particular focus on agent-based models, such as Mueller, M., Bogner, K., Buchmann, T. et al. The effect of structural is parities on knowledge diffusion in networks: an agent-based simulation model. J Econ Interact Coord 12, 613–634 (2017).

Reply:

 We gave a deeper analysis of the existing literature in this field with a particular focus on agent-based models (lines 26-59). We also refined some narrative in this section, and added some references accordingly, including the one given by the reviewer.

---

## [Decision Letter · Decision Letter 2]

6 Jun 2022

Impact of network density on the efficiency of innovation networks: An agent-based simulation study

PONE-D-21-15597R2

Dear Dr. Hua,

We’re pleased to inform you that your manuscript has been judged scientifically suitable for publication and will be formally accepted for publication once it meets all outstanding technical requirements.

Kind regards,

Alessandro Rizzo

Academic Editor

PLOS ONE

Additional Editor Comments (optional):

Reviewers' comments:

Reviewer's Responses to Questions

**Comments to the Author**

1. If the authors have adequately addressed your comments raised in a previous round of review and you feel that this manuscript is now acceptable for publication, you may indicate that here to bypass the “Comments to the Author” section, enter your conflict of interest statement in the “Confidential to Editor” section, and submit your "Accept" recommendation.

Reviewer #2: All comments have been addressed

Reviewer #3: All comments have been addressed

2. Is the manuscript technically sound, and do the data support the conclusions?

Reviewer #2: Yes

Reviewer #3: Yes

3. Has the statistical analysis been performed appropriately and rigorously? 

Reviewer #2: N/A

Reviewer #3: N/A

4. Have the authors made all data underlying the findings in their manuscript fully available?

Reviewer #2: Yes

Reviewer #3: Yes

5. Is the manuscript presented in an intelligible fashion and written in standard English?

Reviewer #2: Yes

Reviewer #3: Yes

6. Review Comments to the Author

Reviewer #2: (No Response)

Reviewer #3: (No Response)

7. PLOS authors have the option to publish the peer review history of their article (what does this mean?). If published, this will include your full peer review and any attached files.

Reviewer #2: **Yes: **Luca Gallo

Reviewer #3: No

---

## [Editor Report · Acceptance letter]

9 Jun 2022

PONE-D-21-15597R2 

Impact of network density on the efficiency of innovation networks: An agent-based simulation study 

Dear Dr. Hua:

I'm pleased to inform you that your manuscript has been deemed suitable for publication in PLOS ONE. Congratulations! Your manuscript is now with our production department. 

Kind regards, 

on behalf of

Prof. Alessandro Rizzo 

Academic Editor

PLOS ONE